# A Concise Paradigm on Radical Hysterectomy: The Comprehensive Anatomy of Parametrium, Paracolpium and the Pelvic Autonomic Nerve System and Its Surgical Implication

**DOI:** 10.3390/cancers12071839

**Published:** 2020-07-08

**Authors:** Mustafa Zelal Muallem, Thomas Jöns, Nadja Seidel, Jalid Sehouli, Yasser Diab, Denis Querleu

**Affiliations:** 1Department of Gynecology with Center for Oncological Surgery, Charité—Universitätsmedizin Berlin, Corporate Member of Freie Universität Berlin, Humboldt-Universität zu Berlin, and Berlin Institute of Health, Berlin, Virchow Campus Clinic, Charité Medical University, 13353 Berlin, Germany; Jalid.sehouli@charite.de; 2Department of Anatomy, Mitte Campus Clinic, Charité Medical University, 10117 Berlin, Germany; Thomas.joens@charite.de (T.J.); Nadja.seidel@charite.de (N.S.); 3Department of Gynecology, Portland Hospital, Portland, VIC 3305, Australia; yaser_diab@yahoo.com; 4Department of Surgery, Institut Bergonié, 33076 Bordeaux, France; denis.querleu@esgo.org

**Keywords:** nerve-sparing radical hysterectomy, parametrium, paracolpium, pelvic autonomic nerve system, tailoring the radicality

## Abstract

The current understanding of radical hysterectomy is more centered on the uterus and little is discussed regarding the resection of the vaginal cuff and the paracolpium as an essential part of this procedure. The anatomic dissections of two fresh and 17 formalin-fixed female pelvis cadavers were utilized to understand and decipher the anatomy of the pelvic autonomic nerve system (PANS) and its connections to the surrounding anatomical structures, especially the paracolpium. The study mandates the recognition of the three-dimensional (3D) anatomic template of the parametrium and paracolpium and provides herewith an enhanced scope during a nerve-sparing radical hysterectomy procedure by precise description of the paracolpium and its close anatomical relationships to the components of the PANS. This enables the medical fraternity to distinguish between direct infiltration of the paracolpium, where the nerve sparing technique is no longer possible, and the affected lymph node in the paracolpium, where nerve sparing is still an option. This study gives rise to a tailored surgical option that allows for abandoning the resection of the paracolpium by FIGO stage IB1, where less than 2 cm vaginal vault resection is demanded.

## 1. Introduction

While the radical hysterectomy, first described by Ernst Wertheim [1], has been in routine medical practice as a standard therapy for locally invasive cervical cancer for more than a century, many discrepancies still persist concerning the terminology, anatomical description and surgical technique. Surgical procedures have been dealing with the parametrium (web, paracervix), a term used to describe the connective tissue around the cervix, for quite a long period as being composed of three ligaments (viz., cardinal ligament, sacrouterine ligament and vesicouterine ligament). Moreover, there are no supporting structures in the parametrium [2,3,4] to rationalize the use of the term ligament. The composition of the parametrium is controversial and nevertheless an issue of anatomic confusion and surgical misconception [5,6]. However, the resection of these parametrial ligaments is a surgical standard and indispensable during radical hysterectomy [7,8]. In 2017, Querleu et al. [9] took up an initiative to standardize the anatomic nomenclature and redefine the technical descriptions of various radical hysterectomy procedures. Nevertheless, specific published literature [5,10,11] reveal that there is still a need for a comprehensive and precise anatomy of radical hysterectomy with extensive knowledge of the pelvic spaces, blood supplies, autonomic nerve system and connective pillars to enable the surgical procedure to be carried out accurately with the right balance as regards radicality, prognosis and quality of life.

A cadaver study was undertaken in addition to our long experience with nerve-sparing radical hysterectomy [12] to arrive at a comprehensive and precise anatomy of the parametrium, paracolpium and the pelvic autonomic nervous system and its connections to surrounding anatomical structures.

## 2. Results

### 2.1. In-Depth Literature Analysis of the Parametrium and Paracolpium

The current understanding of radical hysterectomy is more centered on the uterus, especially the uterine cervix, and little or nothing is discussed on the significance of the resection of the vaginal cuff (which generally runs up to one third or half of the vagina) and the paracolpium as an essential part of a radical hysterectomy [8,9,13]. Since the radical resection of any organ obviously implies the dissection and resection of the vascular supply, lymph vessels and surrounding lymph nodes, there is no published literature available on radical hysterectomies that details the dissection and resection of vaginal vessels or offers a precise description of the paracolpium. Fujii and Sekiyama [14] have described the paracolpium as lateral vaginal tissue (an artery and a vein) that could be distinguished during the final crucial steps of a radical hysterectomy directly before dividing the vagina without clearing where these vessels arise from. However, detailed description of the paracolpium is not lucid in spite of their anatomical positioning in the vesicouterine and vesicovaginal ligament.

Further, the ventral and lateral parametrium (and of course the paracolpium) have extensive vascularization for the blood supply of the uterus (and the upper vaginal region as well), populated with the numerous local lymph nodes and lymphatic supply. It is worth mentioning the contribution of Girardi [15] and Benedetti [16] who have lucidly shown that 78% to 96% of the parametrium contains lymph nodes. Palfalvi and Ungar [17] and Querleu et al. [18] have further elucidated that the border between parametrial dissection and lymphadenectomy is not rigid and could be positioned in different planes, while the same connective tissue is being extirpated. Thus, according to these studies and our experience, the ventral and lateral parametrium and paracolpium are only the vasculatures that run in these structures to supply blood to the uterus and upper vagina, and their surrounding lymph nodes (Figure 1). By dissecting the surrounding lymph nodes, the parametrium and paracolpium resection will only mean the resection of the uterine vessels and the vessels supplying the upper part (one third to one half) of the vagina.

### 2.2. The New Three-Dimensional Precise Anatomic Description of the Parametrium and Paracolpium as a Result of Our Anatomical and Surgical Studies

The cadaver studies and their interpretation in surgical steps primarily mandate the recognition of the three-dimensional (3D) anatomic template proposed as early as 2011 [19] for parametrectomy, and at the same time, a need for an addition of a parallel 3D anatomic template for paracolpectomy too. The dissection has shown the dorsal parametrium as the sacrouterine ligament and the dorsal paracolpium as the sacrovaginal ligament that has been previously described as the deep uterosacral ligament by Ramanah et al. [6], and as the vaginorectal ligament by various other authors [5].

A precise dissection of the area reveals that whilst there is no clear border between the stated structures, the dorsal paracolpium is denser, connects the vaginal part beneath the cervix to the lateral mesorectum and is located in the same level in between the rectal fascia and the pelvic nerve-vessel guiding plate laterally (this plate has also been referred to as the lateral sheet of the presacral visceral pelvic fascia [20]). Therefore, the sacrovaginal ligament is located with a part that is closely connected to the hypogastic nerves and the dorsal part of the inferior hypogastric plexus. The sacrovaginal ligament is connected to the endopelvic fascia (fascia pelvis visceralis) and it would be also appropriate to term it the tendinous arch of the pelvic fascia (arcus tendineus fasciae pelvis) located dorsally at the level of S2/3 [2]. It is essential to differentiate between the tendineus arch of the pelvic fascia the tendineus arch of levator ani muscle.

The lateral paracolpium is where the vaginal vascular supply originates from (artery) and discharges into (vein) the internal iliac artery and vein beneath the ureter.

In this way, it is possible to label the ureter as an anatomical marker that splits the lateral parametrium (cardinal ligament) into the lateral parametrium above the ureter, containing the uterine artery and vein, and the lateral paracolpium beneath the ureter, containing the vaginal artery and vein (Figure 2). These two vessels have been misidentified as a deep uterine vein by some researchers, whilst there is no mention of this deep uterine vein in the available published literature or textbooks on human anatomy. Thus, this vaginal vein as per this study (misinterpreted as the deep uterine vein) originates at the level of the superior third of the vagina [20].

However, it is imperative to mention that the vaginal artery goes undetected laterally in about 14/33 hemi-pelvises in a cadaver study (42.4%) and in 37/84 hemi-pelvises in a clinical study (44%). This is due to the fact that this artery in such instances is construed to come in as a branch of the uterine artery, crossing directly into the ventral parametrium above the ureter and travelling caudally to the anterolateral side of the vagina. Both of these variations could be noticed in the same patient in 23/42 patients from a clinical study (55%) [21] and in 16/28 cadavers with eligible bilateral hemi-pelvises (57.1%). This finding falls in tandem with the investigations of Ercoli et al. [20] in large numbers of embalmed and fresh cadavers.

Hence, the ventral paracolpium is nothing but the deep layer of the vesicouterine ligament (lateral and caudal portions from the distal ureter in the vesicouterine ligament, described in Yabuki et al. [7] as the paravaginal portion of the paracervix) and the superficial layer of the vesicouterine ligament is actually the ventral parametrium (cranial and medial portions from the distal ureter).

Thus, in our dissection, the following vessels: a ureteral branch of the uterine artery, a ureteral vein discharging in the uterine vein (described by Fujii as a superficial vesical vein [22]) and the vaginal branch of the uterine artery (a variation of vaginal artery arising from the internal iliac artery in 56–57.6% of cases) have been clearly identified in the ventral parametrium. In the ventral paracolpium, two veins have been spotted discharging into the vaginal vein and making vein anastomoses with branches from the inferior vesical vein. These are the same two veins described as the middle and deep vesical vein by Fujii [22]. However the present detailed cadaver dissection study and our surgical experience mandates describing them as the lateral vesicovaginal vein (instead of the middle vesical vein) and the medial vesicovaginal vein (instead of the deep vesical vein) because they are anastomotic connections between the vaginal vein and inferior vesical vein (Figure 3). These vessel anastomoses could be explained as the vesical venous plexus that envelops the lower part of the bladder and the upper part of the vagina. The vaginal artery crosses over the inferior hypogastric plexus from the dorsolateral side to the ventromedial side to continue at the ventrolateral sidewall of the vagina.

### 2.3. The Pelvic Autonomic Nervous System and Its Course Across the Paracolpium

The pelvic splanchnic nerves have been found to run directly from the sacral roots in front of the common trunk of the internal pudendal and inferior gluteal vessels at the dorsal edge of the lesser sciatic foramen, and then medial from the inferior vesical vein cranially to merge in the inferior hypogastric plexus.

The inferior hypogastric plexus lies on the endopelvic fascia at the lateral vaginal and rectal sidewall. The bladder branches of the inferior hypogastric plexus leave the plexus medial from the medial vesicovaginal vein ca. 2 cm beneath the vaginal vault (Figure 4).

This study also shows that most of the injuries to the inferior hypogastric plexus or the components of the autonomic nerve system in the radical hysterectomy procedure occur during the resection of the vaginal vault. This element has been reported and substantiated by published literature, which conclude that the more extensive the vaginal and surrounding tissue ablation, the greater the resultant bladder denervation [23]. This supposition is further supported by the finding that voiding dysfunction after a radical hysterectomy is significantly more common in cases where more than 2 cm of the vagina has been ablated [20].

Therefore, it is essential to emphasize the accurate identification of all the components of the autonomic nerve system, especially while attempting a nerve-sparing radical hysterectomy procedure, complying with only the resection of the uterine branches and the dissection of the endopelvic fascia directly at the level of the first bladder branch of the inferior hypogastic plexus [11,12,14] (Figure 5).

### 2.4. The Surgical Impact of the Comprehensive Anatomy of Paracolpium and the Pelvic Autonomic Nervous System

During a radical hysterectomy, the resection of vaginal length that is deemed appropriate for the disease stage has to be calibrated according to the tumor size and infiltration into the vagina. Furthermore, cutting the vaginal cuff without highlighting and isolating the endopelvic fascia (at the pelvic nerve-vessel guiding plate) would lead to injuries of the inferior hypogastric plexus from its medial facets [11]. The endopelvic fascia (pubocervical fascia) runs from the lower part of the symphysis pubis around the bladder neck directly above the pelvic floor (levator ani muscle) to the lateral side of the middle vagina and the lateral side of the rectum. The part from the endopelvic fascia that goes into the rectum is the dorsal parametrium and paracolpium (generally described as the sacrouterine and sacrovaginal ligaments). These two parts are again connected with the superior fascia of the levator ani at the tendinous arch of the pelvic fasciae [24].

Lateral and cranial from this endopelvic fascia is the vascular supply of the vagina and cervix, consisting of the lateral and ventral paracolpium and parametrium. The vaginal vessels cross over the inferior hypogastric plexus to run lateral to the vaginal sidewall; therefore, the last step of a radical hysterectomy will be the resection of the descending vaginal vessels at the lateral side of the vagina. The inferior vesical artery arises from the common trunk of the internal pudendal artery and the inferior gluteal artery (sometimes directly from the internal iliac artery as its last medial branch) and the inferior vesical vein discharges into the internal iliac vein. The inferior vesical vessels are chiefly engaged in supplying the lower wall of the retrovesical ureter [25,26].

## 3. Discussion

The implications of this comprehensive anatomy identification of the parametrium and paracolpium have a fundamental relevance not only in the nerve-sparing radical hysterectomy procedure, but more so with the empowerment of the surgeon to modify the radical hysterectomy according to the tumor size and infiltration. The study outcomes provide an enhanced scope during a nerve-sparing radical hysterectomy procedure by precise identification, isolation and sparing of the inferior vesical vessels to avoid ureteral ischemia at the distal part of the ureter, which usually is prevalent in 6.4% of cases after a radical hysterectomy procedure [1].

Since the ventral and lateral paracolpium are closely connected to the pelvic autonomic nerve system and the vaginal vessels pretty close to the pelvic splanchnic nerves running above the pelvic nerve-vessel guiding plate medially along the lateral vaginal sidewall (Figure 3), the present study enables the medical fraternity to distinguish between direct (continuous) and lymphatic infiltration of the paracolpium, where direct (continuous) infiltration of the paracolpium with involvement of the endopelvic fascia will make it impossible to go for a nerve-sparing technique. On the other hand, if there is only lymphatic spread to the paracolpium (affected lymph node in the paracolpium region or lymph vascular space infiltration in the paracolpium), a nerve-sparing radical hysterectomy can still be performed.

The study also gives rise to a tailored surgical option that allows for abandoning the resection of the paracolpium (ventral, dorsal and lateral) if the tumor is restricted to the cervix, and less than 2 cm in size (demanding less than 2 cm vaginal vault resection).

This will allow for revising the classification of radical hysterectomy [8,9] to redefine type B radical hysterectomy (abandoning the unclear definitions like partial excision) as a radical hysterectomy with resection of the new defined ventral (at the vesical trigone), lateral (at the internal iliac vessels) and dorsal parametrium (at the rectum sidewall) but without paracolpium resection and with minimal (less than 2 cm) vaginal vault, which will be the surgery of choice in stages IA and IB1 (FIGO -International Federation of Gynecology and Obstetrics- 2018 [27]).

The type C radical hysterectomy will then be the radical hysterectomy with parametrium and paracolpium resection for stage IB1 (the old FIGO classification [28]) with deep stromal invasion and IB2-IIA or early IIB cervical cancers. This procedure could be performed nerve sparingly (C1) if there is no direct infiltration in the paracolpium and tendinous arch of the pelvic fascia (endopelvic fascia), or without sparing the inferior hypogastric plexus (C2) only by the direct infiltration of the paracolpium and/or the tendinous arch of the pelvic fascia. Tumor size plays no role whilst making the decision about sparing the autonomic nerve system, which has to be the standard of care for even a >4 cm size tumor [12].

## 4. Materials and Methods

The anatomic dissections of 2 fresh and 17 formalin-fixed female pelvis cadavers were utilized to study, understand and decipher the hitherto ambiguously annotated anatomy of the pelvic autonomic nerve system and its connections to the surrounding anatomical structures, especially the paracolpium, with reference to radical hysterectomy. Since medical students had previously dissected the formalin-fixed cadavers, we excluded five hemi-pelvises where the pelvic sidewall was grossly over-dissected.

The new anatomical knowledge from a cadaver study used to enhance and develop the Muallem technique for nerve-sparing radical hysterectomy was described in a previous study [12]. All data was obtained in accordance with Charité guidelines for cadaveric use in research. The findings are extrapolated by applying strategies to navigate these structures during a live nerve-sparing radical hysterectomy using ultrasonic liposuction (SonoSurg; Olympus Deutschland GmbH, Hamburg, Germany) for the visual illustration of the pelvic autonomic nervous system with less aggressive dissection. Pelvic intraoperative neuromonitoring (pIOM) was used to support the nerve-sparing radical hysterectomy and to confirm the intraoperative functional integrity of the inferior hypogastric plexus. The modular technology consisted of an ISIS Xpert neuromonitoring system including a neurostimulator and multichannel amplifiers for electromyographic recording (EMG). The NeuroExplorer operating software included the pIOM application, providing simultaneous stimulation and recording with the help of a hand-guided disposable 400 ball-tip stimulation probe and needle electrodes (inomed Medizintechnik GmbH, Emmendingen, Germany). The needle electrodes were placed in the internal and external sphincter.

At the end of this study, PubMed and PMC (PubMed Central) were searched for studies pertaining to radical hysterectomy, radical hysterectomy and anatomic complications, anatomy of the pelvic autonomic nerve system and its connections to surrounding anatomical structures, especially the paracolpium and parametrium, and surgical challenges in radical hysterectomy for a complete literature search for comparison and reference.

## 5. Conclusions

The comprehensive anatomy of the parametrium, paracolpium and pelvic autonomic nerve system as presented in this study is a contradictory concise paradigm for radical hysterectomy, which provides a precise concept of the nerve-sparing radical hysterectomy that shall enable surgeons to perform such a complex procedure without the ensuing common ureter related complications. The key steps are to distinguish between direct (continuous) and lymphatic infiltration of the paracolpium, rethink the needed radicality and devise tailored surgical procedures to suit individual cases.

## Figures and Tables

**Figure 1 cancers-12-01839-f001:**
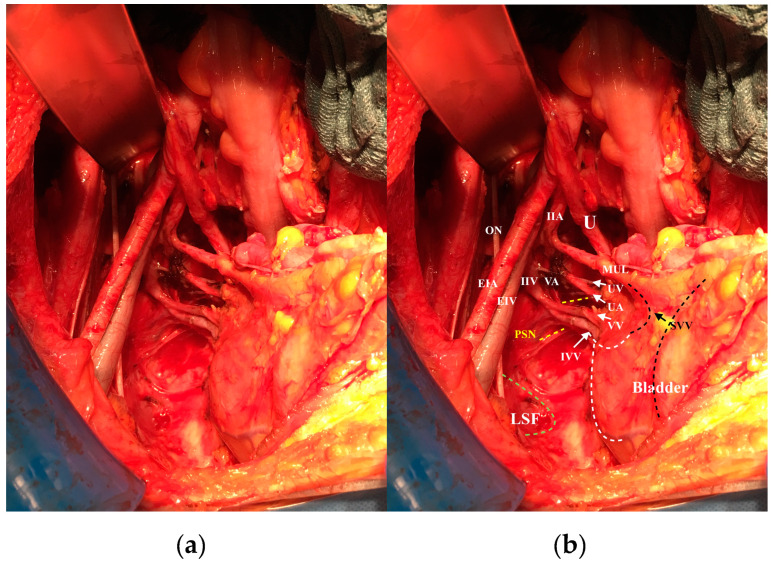
The lateral parametrium and paracolpium after removing the lymph nodes and fat tissue: (**a**) original photo; (**b**) with explanation of anatomic structures: U: ureter, MUL: medial umbilical ligament, IIA: internal iliac artery, IIV internal iliac vein, EIA: external iliac artery, EIV: external iliac vein, ON, obturator nerve, IVV: inferior vesical vessels (the dotted white line shows the distribution of these vessels in the bladder wall), SVV: superior vesical vessels (dotted black line shows the distribution of these vessels in the bladder wall), UV: uterine vein, UA: uterine artery, VV: vaginal vein, VA: vaginal artery, PSN: pelvic splanchnic nerves, LSF: lesser sciatic foramen.

**Figure 2 cancers-12-01839-f002:**
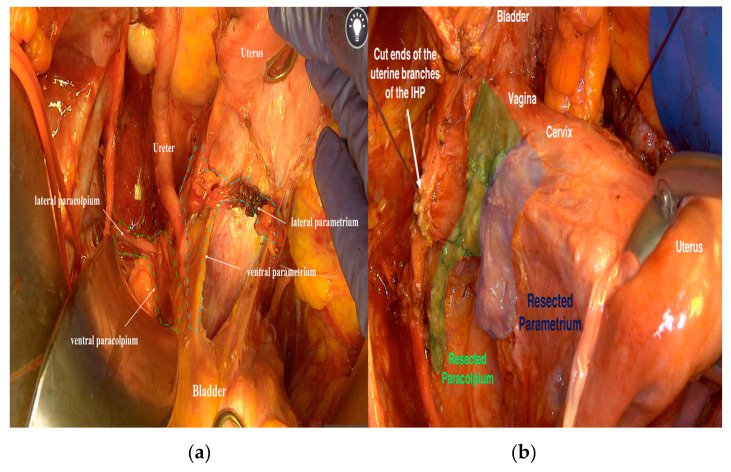
The lateral and ventral parametrium and paracolpium. (**a**) The ureter as an anatomical marker between the lateral parametrium and the lateral paracolpium and between the ventral parametrium and the ventral paracolpium. (**b**) The parametrium and paracolpium after resecting and sparing the pelvic autonomic nerve system. IHP: inferior hypogastric plexus.

**Figure 3 cancers-12-01839-f003:**
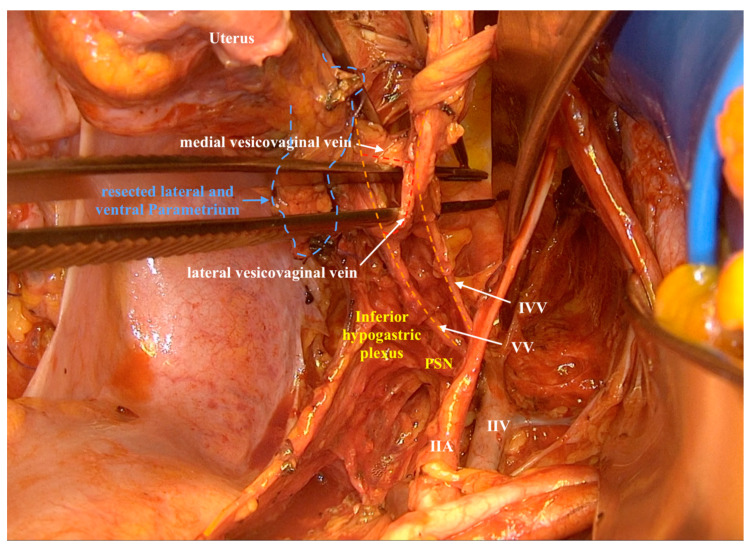
The lateral and ventral paracolpium with the vesicovaginal veins in the ventral paracolpium and their connection to the inferior vesical vein. VV: vaginal vein, IVV: inferior vesical vein, IIA: internal iliac artery, IIV internal iliac vein, PSN: pelvic splanchnic nerves.

**Figure 4 cancers-12-01839-f004:**
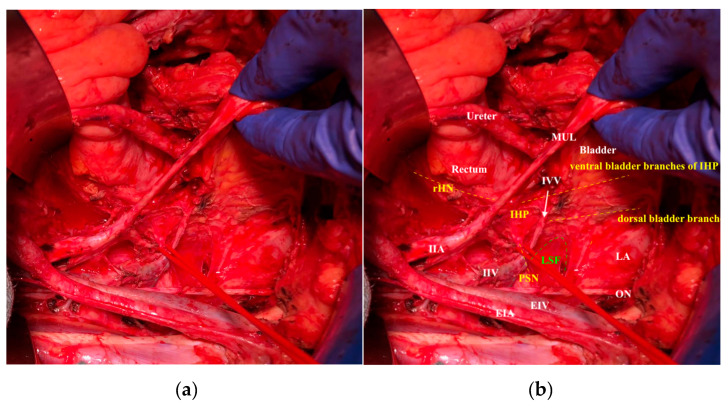
The lateral appearance of the right inferior hypogastric plexus after nerve-sparing radical hysterectomy: (**a**) original photo; (**b**) with explanation of the anatomic structures. MUL: medial umbilical ligament, IIA: internal iliac artery, IIV internal iliac vein, EIA: external iliac artery, EIV: external iliac vein, ON, obturator nerve, IVV: inferior vesical vessels, PSN: pelvic splanchnic nerves, LSF: lesser sciatic foramen, LA: levator ani, IHP: inferior hypogastric plexus, rHN: right hypogastric nerve.

**Figure 5 cancers-12-01839-f005:**
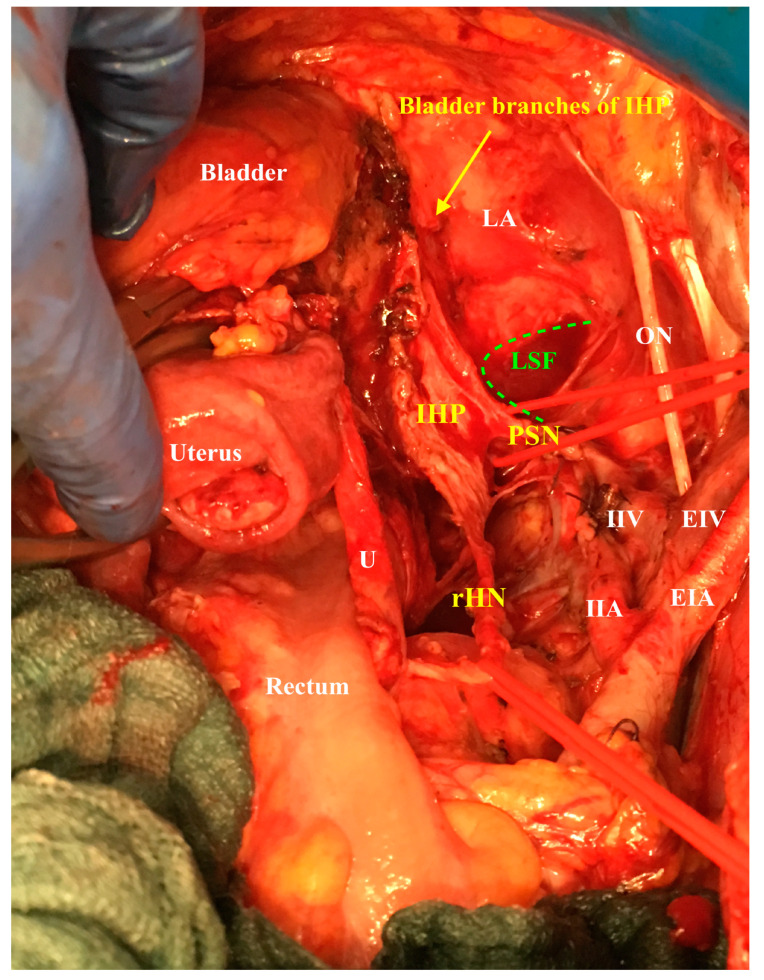
Cranial appearance of the right inferior hypogastric plexus during nerve-sparing radical hysterectomy. U: ureter, IIA: internal iliac artery, IIV internal iliac vein, EIA: external iliac artery, EIV: external iliac vein, ON, obturator nerve, IVV: inferior vesical vessels, PSN: pelvic splanchnic nerves, LSF: lesser sciatic foramen, LA: levator ani, IHP: inferior hypogastric plexus, rHN: right hypogastric nerve.

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
