# Peer review of "A Concise Paradigm on Radical Hysterectomy: The Comprehensive Anatomy of Parametrium, Paracolpium and the Pelvic Autonomic Nerve System and Its Surgical Implication"

_cancers, 2020, doi:10.3390/cancers12071839_

Round 1

Reviewer 1 Report

In the present manuscript "A Concise Paradigm on Radical Hysterectomy:  The comprehensive anatomy of parametrium,  paracolpium and the pelvic autonomic nerve system  and its surgical implication" a detailed and critical description of the anatomical organization of the parametrium, paracolpium and pelvic autonomic nervous system is given, with the intention of providing surgeons with the tools necessary to deal with the complex procedure of radical hysterectomy.

The work is well written and the anatomical study detailed satisfactorily and worthy of pubblication.

I believe this study is extremely important for clinical aspects and provides the anatomical tools to allow surgeons to avoid post-operative complications and to improve the quality of life of the patient who undergoes such frequent surgical practice.

Author Response

Dear reviewer,

thank you very much for reviewing our manuscript and of course for your encouraging comments. Your comments reflect a deep understanding of the point of view of our manuscript.

Best regards.

Reviewer 2 Report

The authors have conducted a study to evaluate the anatomic structure of radical hysterectomy, specifically focusing on the parametrium and paracolpium area. In general, the study is interesting, since the main goal of the successful surgery for cervical cancer is complete and adequate resection. However, the degree of adequate and complete surgery is still debated, although the classification of the types for radical hysterectomy was varied. One of the authors: Denis Querleu has published updated information on the Querleu-Marrow classification of radical hysterectomy in 2017 (Ann Surg Oncol 2017;24:3406-3412.). In addition, the worse outcome using minimally invasive surgery has been reported and well discussed. Therefore, much more clear demonstration of the parametrium structure might be urgently needed. Using three-dimensional anatomic template to identify the parametrial and paracolpium might be a good tool to provide a better reference of the adequate resection. Furthermore, post-radical hysterectomy often results in the dysfunction of the urinary bladder and/or rectum, which results a new technological development-nerve-sparing radical hysterectomy. Although the recent reports have shown that the patients treated with near-sparing radical hysterectomy might have the similar survival outcome compared to those treated with standard radical hysterectomy, and this finding was also noted in the previous studies (retrospective, cohort study, or case series) before the 2018 publication of LACC which was conducted to compare Minimally Invasive and Abdominal Radical Hysterectomy for Cervical Cancer. After this publication, nearly all studies were against the use of minimally invasive radical hysterectomy in the management of the patients with cervical cancer. Therefore, the current study might provide the much more basic understandings of the spreading pattern of the cervical cancer using the modern technology. I personally favor this article. However, I am still confused the goal of the current article. Did the authors recommend that all patients should be evaluated by this strategy before a surgical plan for cervical cancer? In addition, it is unknown the degree of resection of parametrium and/or paracolpium. In fact, parametrial tissue can be easily separated into three parts from the corpus, cervix, to vaginal side (Int J Gynaecol Obstet. 2003 Feb;80(2):145-51.). In addition, the question “is more radical more effective?” is raised, since the recent trend has attempted to use multi-modality in the management of local advanced diseases with maximal effort to preserve or maintain the function or organs. Could the authors kindly comment this part?

Furthermore, unlikely to epithelial ovarian cancer (Ro surgery without any residual tumor provides the best outcome compared to other degree of the debulking surgery) it is hard to identify how extent of resection for parametrium or paracolpium can be related to the outcome of cervical cancer patients. Without evidence to show it, the argument may be continuous.

Moreover, surgery is not of choice of treatment for cervical cancer, although concurrent chemoradiation, or new targeted therapy has been continuously investigated, if the latter can overcome the dysfunction of pelvic organ.  

Finally, there is still absent of idea tool which could really detect the concealed or extremely microscopic spreading of the tumor, contributing to question the possibility of over-enthusiasm of the authors’ claims.

Author Response

Dear reviewer,

Thank you very much for reviewing our manuscript and for your very valuable comments, which reflect your deep understanding of the surgical point of views about cervical cancer. As you noticed, we try with this manuscript and the other following ones to address these subjects in an attempt to achieve a consensus about radical hysterectomy (indication, best technique taking into account the postoperative quality of life, and tailoring the surgery to avoid over or under therapies).

This paper aims to describe a comprehensive and precise anatomy of radical hysterectomy with extensive knowledge of the pelvic spaces, blood supplies, autonomic nerve system and connective pillars to enable the surgical procedure to carry out accurately with the right balance as regards radicality, prognosis and quality of life.

The implications of this comprehensive anatomy of parametrium and paracolpium have a fundamental relevance not only in the nerve sparing radical hysterectomy procedure, but more so with the empowerment of the surgeon to modify the radical hysterectomy according to the tumor size and infiltration. We think that we need a new classification for radical hysterectomy, which gives us the ability to classify the procedure according to the radicality needed for adequate resection and not according to the difficultness of operation! In this term, we support the multimodality in treatment of advanced cervical cancer, and believe that the surgery still has a very important role to play even by locally advanced tumor.

In this paper, we do not describe a preoperative strategy for a surgical plan by cervical cancer, but we highlight the precise anatomy of paracolpium and pelvic autonomic nerve system, giving the chance of an enhanced scope during a radical hysterectomy to be able to avoid complications (like ureter ischemia) and to spare the inferior hypogastric plexus components. In this term, the study does not encourage the surgeon to be more radical but it gives them the map to be able to navigate the right ways for nerve-sparing radical hysterectomy, for adequate resection and even perhaps for tailoring the surgery in small tumors.

We found, that parametrium and paracolpium are nothing else more than the blood supply of uterus (cervix) and upper vagina with the combined lymph nodes and lymph ways. In this way, we think that the partial resection of parametrium or paracolpium has no meaning. The tailoring of surgery has to be indicated based on the indication to resect vaginal vault for more than 2 cm or not and of course then to resect the paracolpium or not.

Finally, we hope that you will support the publishing of this paper as a step on the long road to be able to enhance our surgical practice in cervical cancer by better understanding of the precise anatomy of paracolpium, parametrium and pelvic autonomic nerve system.

Best regards

Reviewer 3 Report

This is an interesting study with very detailed anatomical description, performed by an expert team in the field. 

I have two comments:

  • the quality of pictures could be improved, specially in figures 2 and 4 (figure 2: unapropriate format, figure 4: unapropriate contrast and colors)
  • the impact of the study on surgical practices should be further explained. The anatomical descriptions are very detailed and are not necessarily accessible and applicable for the vast majority of surgeons and the emphasis should be on surgical training and the way in which the authors envisage that the surgeons could appropriate their anatomical classification.

Author Response

Dear reviewer,

Thank you very much for reviewing our manuscript and for your very valuable comments, which reflect your deep understanding of the surgical point of views about cervical cancer.

We will upload the same photos with better resolutions.

Your second point is very important and we share you the same opinion that the surgical training has to be emphasized by discussing a complex operation technique like nerve-sparing radical hysterectomy. In this manuscript, we described our results after anatomical study. In our next paper, we will concern more on surgical steps, learning curves and the importance of surgical training.

On the other hand, we think that the highlighting of this precise anatomy of parametrium and paracolpium is very essential not only in the nerve sparing radical hysterectomy procedure, but more so with the empowerment of the surgeon to modify the radical hysterectomy according to the tumor size and infiltration.

Finally, we hope that you will support the publishing of this paper as a step on the long road to be able to enhance our surgical practice in cervical cancer by better understanding of the precise anatomy of paracolpium, parametrium and pelvic autonomic nerve system.

Best regards